# Microsatellite Status Detection in Gastrointestinal Cancers: PCR/NGS Is Mandatory in Negative/Patchy MMR Immunohistochemistry

**DOI:** 10.3390/cancers14092204

**Published:** 2022-04-28

**Authors:** Federica Zito Marino, Martina Amato, Andrea Ronchi, Iacopo Panarese, Franca Ferraraccio, Ferdinando De Vita, Giuseppe Tirino, Erika Martinelli, Teresa Troiani, Gaetano Facchini, Felice Pirozzi, Michele Perrotta, Pasquale Incoronato, Raffaele Addeo, Francesco Selvaggi, Francesco Saverio Lucido, Michele Caraglia, Giovanni Savarese, Roberto Sirica, Marika Casillo, Eva Lieto, Annamaria Auricchio, Francesca Cardella, Ludovico Docimo, Gennaro Galizia, Renato Franco

**Affiliations:** 1Pathology Unit, Department of Mental and Physical Health and Preventive Medicine, University of Campania “L. Vanvitelli”, 80138 Naples, Italy; federica.zitomarino@unicampania.it (F.Z.M.); martina.amato@unicampania.it (M.A.); andrea.ronchi@unicampania.it (A.R.); iacopo.panarese@unicampania.it (I.P.); franca.ferraraccio@unicampania.it (F.F.); 2Department of Precision Medicine, University of Campania “L. Vanvitelli”, 80138 Naples, Italy; ferdinando.devita@unicampania.it (F.D.V.); giuseppe.tirino@unicampania.it (G.T.); erika.martinelli@unicampania.it (E.M.); teresa.troiani@unicampania.it (T.T.); michele.caraglia@unicampania.it (M.C.); 3Medical Oncology Unit, SM delle Grazie Hospital, 80078 Pozzuoli, Naples, Italy; gaetano.facchini@aslnapoli2nord.it; 4General Surgery Unit, Santa Maria delle Grazie Hospital, 80078 Pozzuoli, Naples, Italy; felice.pirozzi@hotmail.it; 5Hepatology and Interventional Ultrasound Unit, San Giovanni di Dio Hospital, 80027 Frattamaggiore, Naples, Italy; perrottamichele@tiscali.it; 6Medical Oncology Unit, ASL Napoli 2 Nord Hospital, 80014 Giugliano, Naples, Italy; pasquale.incoronato@aslnapoli2nord.it; 7Medical Oncology Unit, San Giovanni di Dio Hospital, 80027 Frattamaggiore, Naples, Italy; raffaele.addeo@aslnapoli2nord.it; 8Department of Advanced Medical and Surgical Sciences, Università degli Studi della Campania Luigi Vanvitelli, 80138 Naples, Italy; francesco.selvaggi@unicampania.it (F.S.); francescosaverio.lucido@unicampania.it (F.S.L.); ludovico.docimo@unicampania.it (L.D.); 9AMES, Centro Polidiagnostico Strumentale srl, 80013 Casalnuovo, Naples, Italy; giovanni.savarese@centroames.it (G.S.); roberto.sirica@centroames.it (R.S.); marika.casillo@centroames.it (M.C.); 10Department of Translational Medical Science, Università degli Studi della Campania Luigi Vanvitelli, 80138 Naples, Italy; eva.lieto@unicampania.it (E.L.); annamaria.auricchio@unicampania.it (A.A.); francescacardella@gmail.com (F.C.); 11Department of Surgical Sciences, School of Medicine, Second University of Naples, Place Miraglia, 3th Building, West Side, 4th Floor, 80138 Naples, Italy; gennaro.galizia@unicampania.it

**Keywords:** microsatellite instability, gastrointestinal cancers, mismatch-repair-system-protein deficient, mismatch-repair-system-protein patchy, PCR

## Abstract

**Simple Summary:**

Microsatellite instability (MSI) detection has a high impact on eligibility for immune checkpoint inhibitors in gastrointestinal cancers. The appropriate detection of MSI represents the major critical issue in clinical practice, thus a better understanding of the limits related to MSI testing is needed to avoid misinterpretations. This study addresses the discordance between IHC and PCR/NGS testing in a large retrospective series of colorectal and gastric cancers in order to improve diagnosis. Our findings show a disagreement between negative/patchy expression IHC and PCR/NGS results, suggesting that molecular testing is mandatory in this subset of tumors.

**Abstract:**

Background: Microsatellite instability (MSI) is a predictive biomarker for immune checkpoint inhibitors. The main goal was to investigate the discordance between IHC and PCR/NGS for MSI testing in gastrointestinal cancers. Methods: Two series were analyzed through IHC for mismatch-repair-system proteins (MMRP) and PCR, with one series of 444 colorectal cancers (CRC) and the other of 176 gastric cancers (GC). All cases with discordant results between IHC and PCR were analyzed by NGS. IHC staining was evaluated as follows: proficient MMR (pMMR), with all MMR positive; deficient MMR (dMMR), with the loss of one heterodimer; and cases with the loss/patchy expression of one MMR (lo-paMMR). Cases with instability in at least two markers by PCR were MSI-high (MSI-H) and with instability in one marker, MSI-low (MSI-L). Cases without instability were evaluated as microsatellite-stable (MSS). Results: In the CRC cohort, 15 out of 444 cases were dMMR and 46 lo-paMMR. Among the 15 dMMR, 13 were MSI-H and 2 MSS. Among the 46 lo-paMMR, 13 were MSI-H and 33 were MSS. In the GC cohort, 13 out of 176 cases were dMMR and 6 cases lo-paMMR. Among the 13 dMMR, 12 were MSI-H and only 1 was MSS. All six lo-paMMR cases were MSS. All NGS results were in agreement with PCR. Conclusions: In clinical practice, MMR–IHC could be used as a screening test and additional molecular analysis is mandatory exclusively in cases carrying loss/patchy MMR-IHC.

## 1. Introduction

The DNA mismatch repair (MMR) complex is a highly conserved DNA repair mechanism that ensures genomic integrity by repairing erroneous short insertions, short deletions and single-base mismatches during DNA replication [1]. DNA mismatch-repair system deficiency results in a significant increase in DNA replication errors that leads to frequent mutations in microsatellite DNA, defined as microsatellite instability (MSI). MSI is a frequent mechanism related to the development of tumors, occurring in almost 95% of Lynch syndrome–associated malignancies, mainly colorectal and endometrial cancers. Moreover, MSI has been reported in 15% of sporadic colorectal cancers (CRCs) and in approximately 10% of gastric cancers (GCs) [1,2,3]. MSI tumors show unique features, defining a subset of patients sensitive to immune-checkpoint inhibitors [4]. MSI detection is mandatory for this therapeutic choice, since two immune-checkpoint inhibitors (ICI), pembrolizumab (Keytruda) and nivolumab (Opdivo), have been approved by the Food and Drug Administration (FDA) for metastatic MSI CRC patients [5]. Recently, the KEYNOTE-177 phase III trial demonstrated the superiority of pembrolizumab in treatment-naive mCRC over first-line chemotherapy with or without biological agents in terms of progression-free survival (PFS), overall response rate (ORR) and safety [6]. At present, MSI in GC could be a negative predictor of the efficacy of adjuvant or neoadjuvant chemotherapy and also a biomarker used for the selection of patients eligible for ICI treatment [7]. Furthermore, post hoc analyses of randomized controlled trials (RCTs) suggested the superior efficacy of anti-PD-1-based therapy compared with standard chemotherapy in MSI-high GC patients [8,9,10,11,12,13]. Based on these data, immunotherapy could be routinely used as the treatment of choice for MSI-high CRCs and GCs patients. MSI status is currently defined by immunohistochemistry (IHC) and DNA molecular testing, mainly by polymerase chain reaction (PCR) and most recently also next generation sequencing (NGS) [14,15]. The IHC approach is based on the detection of MMR protein expression, including MLH1, MSH2, MSH6 and PMS2. MMR function requires protein heterodimerization as follows: MSH2 combines with MSH6 and MLH1 combines with PMS2. MSH2 and MLH1 play a pivotal role since their loss leads to the degradation of the partners in their respective heterodimers [16]. The defective expression of mismatch-repair proteins (dMMR) represents a surrogate for the MSI phenotype, while a proficient MMR (pMMR) corresponds to microsatellite stability (MSS) [17]. PCR analysis detects the amplified microsatellite loci, and two different panels of consensus nucleotide repeats are currently available, such as the Bethesda and the Pentaplex panels [16,18,19]. Based on PCR results, MSI is classified into high microsatellite instability (MSI-H) if at least two markers are unstable and low microsatellite instability (MSI-L) if only one marker is unstable [20]. Previous studies reported discordance between MMR-IHC and MSI-PCR results, ranging from 1% to 10% in the CRC series [16,21,22,23,24,25]. Few data have been reported about MSI testing in GC [26,27]. The appropriate detection of MSI represents the major critical issue in the selection of patients for ICI treatment, thus a better understanding of the pitfalls in MSI testing is needed to avoid misinterpretations. This study addresses the discordance between MMR-IHC and MSI-PCR testing in a large retrospective series of CRCs and GCs, in order to clarify the discrepancies and improve diagnosis aimed at treatment with specific inhibitors.

## 2. Materials and Methods

### 2.1. Specimens

#### 2.1.1. Series of Colorectal Cancers

A series of 444 colorectal cancer tumor tissue samples from surgical resections or biopsies performed between 2019 and 2021 at the University of Campania “L. Vanvitelli”, the Santa Maria delle Grazie Hospital and the ASL Napoli 2 Nord Hospital were collected. Among 444 cases selected, 384 were surgical samples and 60 biopsies. All 444 cases were reviewed according to the current histological WHO classification. We retrospectively recorded clinical and pathological parameters, including age of the patient at initial diagnosis, gender, tumor site, histological type, grade, B-RAF status, K-RAS status and N-RAS status.

#### 2.1.2. Series of Gastric Cancers

A series of 176 gastric cancer tumor tissue samples from surgical resections or biopsies performed between 2019 and 2021 at the University of Campania “L. Vanvitelli”, the Santa Maria delle Grazie Hospital and the ASL Napoli 2 Nord Hospital were collected. Among the 176 cases analyzed, 85 were surgical samples and 91 biopsies. All 176 cases were reviewed according to the Lauren classification. We retrospectively recorded clinical and pathological parameters, including age of the patient at initial diagnosis, gender, tumor site, Lauren type, grade, HER2 status, EBV status and PD-L1 status.

### 2.2. Immunohistochemical Analysis of MMR Protein Expression

Immunohistochemical analysis (IHC) was performed on 4 µm thick whole sections for each case. MMR-IHC was carried out using four antibodies directed against MLH1 (M1 Ventana clone ready for use and Optiview kit revelation, Tucson, AZ, USA), MSH2 (clone G219-1129 Ventana ready to use; Optiview kit revelation), MSH6 (clone SP93 Ventana ready to use; Optiview kit revelation), and PMS2 (clone A16-4 Ventana ready to use; Optiview kit revelation) proteins on the BenchMark XT device (Ventana Medical Systems). Adjacent normal tissue from each sample served as positive controls. MMR protein loss was defined by the absence of IHC staining in the nucleus of tumor cells while normal cells remained stained, ensuring the technical validity of the experiment. Immunohistochemical staining results were evaluated according to the scoring system reported in the literature: (i) proficient MMR (pMMR), cases showing positive staining of all four MMR; (ii) defective expression of mismatch repair proteins (dMMR), cases carrying the loss of one of two heterodimers, including MLH1/PMS2 or MSH2/MSH6 loss [9]. We further considered another subset: (iii) cases harboring the loss of one MMR and/or the patchy expression of one or more MMR (lo-paMMR). Two independent observers carried out immunohistochemical analysis, and both observers were blinded.

### 2.3. PCR Detection for Microsatellite Status

Serial sections of 6 µm in thickness from formalin-fixed paraffin-embedded matched normal and tumor tissues were routinely stained, and representative normal and tumor regions were identified by microscopic examination. Genomic DNA was isolated from the paraffin-embedded tissues using the QIAamp DNA mini kit (Qiagen, Valencia, CA, USA) following separation of tumor and normal tissue by manual microdissection. MSI was determined on tumor DNA using the EasyPGX^®^ readyMSI, including the following mononucleotide repeats: BAT25, BAT26, NR21, NR22, NR24, NR27, CAT25 and MONO27. The test was performed according to the manufacturer’s instructions. PCR results were evaluated as follows: (i) microsatellite stable (MSS), cases with none of the markers unstable; (ii) microsatellite instability-high (MSI-H), tumor with 2 or more unstable markers; (iii) microsatellite instability-low (MSI-L), cases with only one marker unstable (in these cases new testing was carried out on non-tumor tissue, if available, to define a germinal mutation).

### 2.4. NGS Detection for Microsatellite Status

All cases with discordant results between IHC and PCR were analyzed by NGS, specifically 25 CRC cases and 9 GC cases. Tumor DNA from selected tumors was sequenced using Illumina TruSightTM Oncology 500 (TSO500) for MSI status determination. The library was prepared according to the manufacturer’s protocol using a hybrid capture-based TruSight Oncology 500 DNA/RNA NextSeq Kit (Illumina, San Diego, CA, USA). During library preparation, enrichment chemistry was optimized to capture nucleic acid targets from FFPE tissues. In the TSO 500 analysis, unique molecular identifiers were used to determine the unique coverage at each position and to reduce the background noise caused by sequencing and deamination artifacts in the FFPE samples. The MSI score was calculated using 130 homopolymer microsatellite loci targeted by the TSO500 panel according to the manufacturer’s instructions. The proportion of unstable MSI sites to total assessed MSI sites was reported as a sample-level microsatellite score, in which at least 40 sites were required to determine an MSI score. The MSI status was calculated from microsatellite sites for evidence of instability relative to a set of baseline normal samples that are based on information entropy metrics. The final NGS study results were reviewed by a bioinformatics expert [28].

### 2.5. Statistical Analysis

Statistical analysis was carried out using IBM SPSS statistics 20. The agreement rate between MMR-IHC and MSI-PCR results was calculated. The Cohen’s kappa coefficient (κ) of agreement was estimated as a measure of the agreement. Kappa values from 0 to 0.2, 0.21 to 0.4, 0.41 to 0.6, 0.61 to 0.8 and 0.81 to 1.0 indicate none, minimal, weak, moderate and strong agreement, respectively [29].

## 3. Results

### 3.1. Colon Cancer Series

#### 3.1.1. Clinicopathological Features

In our study, we analyzed 444 cases of CRC. The mean age of patients was 67 years (range 26–95 years), 181 (40.76%) were younger and 263 (59.24%) were older than 67 years of age. In our series, 265 out of 444 (59.68%) were male and 179 (40.32%) female. Tumor location was the ascending colon in 123 (27.71%) cases, transverse colon in 7 (1.57%), descending colon in 117 (26.36%), sigma-rectum in 89 (20.04%), metastasis in 24 (5.40%) cases and not available in 84 (18.92%) cases. Our cohort included 407 (91.67%) adenocarcinomas, 33 (7.21%) mucinous and 4 (1.12%) signet ring cell carcinomas. Of the 444 CRC, 246 were grade II (55.41%), 76 were grade III (17.12%) and 24 were grade I (5.40%) and in 98 cases (22.07%) grade was not available. Among 444 patients analyzed, 326 (73.42%) were BRAF wild-type (wt), 26 (5.86%) were BRAF mutated and in 92 cases (20.72%) BRAF status was not available. Of the 444 cases, 199 (44.82%) were KRAS wt, 153 (34.46%) were KRAS mutated and in 92 cases (20.72%) KRAS status was not available. Of the 444 cases, 338 (76.13%) were NRAS wt, 14 (3.15%) were NRAS mutated and in 92 cases (20.72%) NRAS status was not available. Clinical and pathological features of CRC patients are summarized in Table 1.

Among 444 cases analyzed, 384 were surgical samples and 60 biopsies. Only 10 biopsies had the corresponding surgical specimen that we analyzed.

#### 3.1.2. MMR-IHC Analysis

Of the 444 CRC cases analyzed by IHC, there were 383 cases (86.3%) proficient MMR, 15 cases dMMR (3.4%), 46 cases with loss of one MMR/patchy expression of one or more MMR (lo-paMMR) (10.4%). In our series, MMR-IHC alteration was observed in about 13.7% of the cases analyzed, indicatively in accordance with the percentage reported in The Cancer Genome Atlas Colorectal Adenocarcinoma (TCGA-COADREAD) dataset [30]. Among dMMR cases, 5 (33.33%) out of 15 cases were BRAF mutated, 2 cases (13.33%) were KRAS mutated and no case was NRAS mutated. Among lo-paMMR cases, 9 (19.57%) out of 46 were BRAF mutated, 11 cases (23.91%) were KRAS mutated and no case was NRAS mutated. Clinical and pathological features of CRC patients harboring dMMR and lo-paMMR are summarized in Table 1.

Among 15 cases dMMR, 14 cases were negative for MLH1/PMS2 (Figure 1A) and 1 negative for PMS2/MSH6 (Figure 2A). Among 46 cases lo-paMMR, 4 cases showed the loss of only one MMR and the patchy expression of another MMR; one case showed the loss of MSH6 and the patchy expression of MLH1/PMS2; eleven cases showed the loss of only one MMR (Figure 3A); eight cases showed the patchy expression of two MMR (Figure 4A); and 23 cases showed the patchy expression of only one MMR (Figure 5A). These data are summarized in Table 2.

The vast majority of dMMR and lo-paMMR cases showed loss of PMS2, in 42.6% of cases (26/61 cases), and with PMS2 patchy expression, in 39.3% of cases (24/61 cases).

The PMS2 loss and patchy expression determined the MSI status in a high percentage (88.5%) of cases confirmed as MSI-H by PCR, similar to previous reports [23,31,32,33]. MSH2 and MSH6 account for a much smaller percentage (3.8% and 11.5%, respectively), while MLH1 accounted for 57.7%, similar to previous reports [23,31,32,33]. These data are summarized in Table 2.

Moreover, all 61 cases harboring dMMR or lo-paMMR were analyzed by PD-L1 IHC. Among 61 cases, 7 cases were PD-L1 positive; specifically, 2 cases dMMR, 4 cases with PMS2 negative and 1 case MLH1-PMS2 patchy. All data are summarized in Appendix A. Furthermore, a series of 60 control cases determined as pMMR by IHC was analyzed by PD-L1 IHC, and all cases were negative for PD-L1 expression.

#### 3.1.3. PCR Detection for Microsatellite Status

Of the 444 CRC cases analyzed by IHC, all 61 cases harboring dMMR or lo-paMMR were analyzed by PCR. Among 61 cases, 32 cases were MSS, 26 cases were MSI-H and 3 cases MSI-L. Among 26 cases MSI-H, 12 cases showed the instability of all loci, while 14 showed the instability of at least two loci. In our series, the frequency of different loci instability was as follows: 96.1% for CAT25, 92.3% for BAT26 and NR27, 88.5% for BAT25 and NR21, 69.2% for NR24 and MONO27, 53.85% for NR22. All 3 cases MSI-L showed the instability of the locus NR21, and the healthy counterparts of these cases showed the same alterations suggesting that the instability of the locus NR21 should occur due to germline mutations (Appendix A). Furthermore, we selected 150 out of 383 CRC pMMR IHC and we analyzed these cases by PCR, and all cases confirmed MSS status.

#### 3.1.4. Comparison of MMR-IHC and MSI-PCR

Discordant results between IHC and PCR were obtained in surgical samples of CRC cases harboring dMMR or lo-paMMR. Among 15 cases dMMR by IHC, 13 were confirmed MSI-H (Figure 1B and Figure 2B) and 2 were MSS. The 2 cases dMMR-IHC and MSS-PCR showed the loss of the same heterodimer MLH1-PMS2. The cases lo-paMMR by IHC showed different results by PCR. Among 46 cases lo-paMMR, 30 cases were MSS (Figure 4B), 13 cases MSI-H and 3 cases MSI-L. Among 11 cases carrying loss of one MMR, 8 cases with loss of MSH6 or PMS2 were MSI-H (Figure 3B) and 3 with loss of PMS2 were MSS by PCR (Table 2). Among 31 cases carrying the patchy expression of one or two MMR by IHC, 25 cases were MSS, 3 were MSI-H (Figure 5B) and 3 were MSI-L. Among 4 cases showing the loss of only one MMR and the patchy expression of another MMR by IHC, 2 were MSI-H and 2 were MSS. These data are summarized in Table 2.

The agreement between the MMR-IHC and MSI-PCR in our series results was moderate (k = 0.618, 95% CI, *p* = 0.000).

Moreover, 25 out of 444 cases were further analyzed by NGS, and all cases had contributory results, specifically: 7 cases pMMR-IHC and MSS-PCR confirmed MSS status by NGS; 2 cases dMMR-IHC and MSS-PCR showed MSS status by NGS; 8 cases loMMR-IHC and MSI-H-PCR showed MSI-H status by NGS; 3 cases paMMR-IHC and MSI-H-PCR showed MSI-H status by NGS; 3 cases paMMR-IHC and MSI-L-PCR showed MSS status by NGS; 2 cases lo-paMMR-IHC and MSI-H-PCR showed MSI-H status by NGS. All data are summarized in Appendix A. No discordant results between IHC and PCR were obtained in CRC biopsy specimens. Considering all cases of our CRC series analyzed both by IHC and PCR, the IHC showed a sensibility of 50% and a specificity of 99%.

#### 3.1.5. Second Diagnostic Opinion of IHC and PCR

The second opinion of IHC interpretation did not change the diagnosis in any of 383 cases of pMMR, while diagnostic discrepancies were observed in 6 out of 61 cases of dMMR and lo-paMMR. The second opinion in 3 cases has changed the MMR-IHC diagnosis; specifically, 3 cases defined as lo-paMMR by the first pathologist were redefined as dMMR. Among these 3 cases, the second opinions were in agreement with MSI-H PCR results in 2 out of 3 cases, while the last one case was MSS by PCR. In the remaining 3 cases, although the interpretation of the second pathologist was discordant with the first regarding the definition of one MMR immunohistochemical staining, the second opinion did not change the overall diagnosis of the cases which were confirmed as lo-paMMR (Table 3).

## 4. Gastric Cancer

### 4.1. Gastric Cancer Series

#### 4.1.1. Clinical and Pathological Characteristics of Patients

In our study, we analyzed 176 cases of GC. The mean age of patients was 65 years (range 34–91 years), 95 (53.98%) were younger and 81 (46.02%) were older than 65 years of age. In our series, 117 (66.48%) were male and 59 (33.52%) female. Tumor location was gastro-esophageal junction in 66 (37.49%) cases, antrum-body in 33 (18.19%), body in 61 (34.66%) cases, metastasis in 4 (2.27%) and not available in 13 (7.39%) cases. Our cohort included 124 (70.45%) intestinal type, 45 (25.57%) diffuse type and 7 (3.98%) mixed type according to Lauren classification. Of the 176 GC, 90 (51.14%) were grade III, 32 (18.18%) were grade II, 8 (4.54%) were grade I and the grade was not available in 46 cases (26.14%). Among 176 patients analyzed, 106 (60.23%) were HER2 negative, 17 (9.66%) HER2 positive and the HER2 status was not available in 53 cases (30.11%). In our series, 135 (76.70%) out of 176 cases were EBV negative, only 1 case (0.57%) was EBV positive and the EBV status was not available in 40 cases (22.73%). PD-L1 was negative in 82 (46.59%) out of 176 cases, positive in 43 (24.43%) and not available in 51 cases (28.98%). Clinical and pathological features of GC patients are summarized in Table 4. Among 176 cases analyzed, 85 were surgical samples and 91 biopsies. Only 7 biopsies had the corresponding surgical specimen that we analyzed.

#### 4.1.2. MMR-IHC Analysis

Of the 176 GC cases analyzed by IHC, there were 157 cases (89.2%) proficient MMR, 13 cases dMMR (7.4%), 6 cases with lo-paMMR (3.4%). In our series, MMR-IHC alteration was observed in about 10.8% of the cases analyzed compared to 21% MSI frequency reported in the Cancer Genome Atlas Stomach Adenocarcinoma (TCGA-STAD) dataset. [34] This discrepancy could be justified since the MSI GC subgroup in TCGA classification is mostly due to promoter methylation of MLH1 rather than the mutations of MLH1, MSH2, MSH6 and PMS2 analyzed in our study. Among dMMR cases, 4 (30.77%) out of 13 cases were PD-L1 positive and all cases were HER2 and EBV negative. Among lo-paMMR cases, 2 (33.33%) out of 6 were PD-L1 positive, only 1 case (16.67%) was EBV positive and all cases were HER2 negative. Clinical and pathological features of GC patients harboring dMMR and lo-paMMR are summarized in Table 4.

All 13 cases dMMR showed the negative staining of the heterodimer MLH1/PMS2 (Figure 6A). Among 6 cases lo-paMMR, 1 case showed the loss of PMS2; 2 cases showed the patchy expression of MLH1-PMS2; 3 cases showed the patchy expression of PMS2 (Figure 7A). These data are summarized in Table 4.

#### 4.1.3. PCR Detection for Microsatellite Status

Of the 176 GC cases analyzed by IHC, all 19 cases harboring dMMR or lo-paMMR were analyzed by PCR. Among 19 cases, 12 cases were MSI-H, 6 cases were MSS and only 1 case was MSI-L. Among 12 cases MSI-H, 6 cases showed the instability of all loci, while 6 cases showed the instability at least of two loci. In our series, the frequency of different loci instability was as follows: 100% for BAT26 and NR27; 91.7% for CAT25, MONO27 and BAT25; 66.7% for NR24, 58.3% for NR21 and NR22 (Appendix A). Furthermore, we selected 100 out of 157 GC pMMR IHC and analyzed these cases by PCR, and all cases confirmed MSS status.

#### 4.1.4. Comparison of MMR-IHC and MSI Molecular Testing

In our series, the comparison between MMR-IHC and MSI-PCR showed high agreement. Among 13 cases dMMR by IHC, 12 were confirmed MSI-H and only 1 was MSS (Figure 6B). Among 6 cases harboring lo-paMMR by IHC, 5 cases confirmed MSS status (Figure 7B) and only 1 was MSI-L by PCR testing. These data are summarized in Table 5.

The agreement between the MMR-IHC and MSI-PCR in our series results is strong (k = 0.957, 95% CI, *p* = 0.000).

Moreover, 9 out of 176 cases were further analyzed by NGS, although 4 out of 9 cases did not have contributory results due to inadequate biomaterial for NGS analysis. All 3 cases dMMR-IHC and MSI-H by PCR confirmed the instability by NGS, the case MLH1-PMS2 negative and MSS-PCR confirmed MSS status by NGS and the case PMS2 patchy and MSI-L-PCR showed MSS status by NGS. All data are summarized in Appendix A. No discordant results between IHC and PCR were obtained in GC biopsy specimens. Considering all cases of our GC series analyzed both by IHC and PCR, the IHC showed a sensibility of 100% and a specificity of 99%.

#### 4.1.5. Second Diagnostic Opinion of IHC and PCR

The second opinion of IHC interpretation did not change the diagnosis in any of the 176 GC cases analyzed in our study.

## 5. Discussion

Microsatellite instability detection has a high impact on eligibility for ICI treatment of dMMR tumors, thus optimization of the testing is required in clinical practice.

MSI status detection can be performed through IHC or molecular approach, the first method based on the loss expression of MMRs that are ubiquitously expressed in cell nuclei and the second based on PCR amplification of microsatellite markers using mono- and di- nucleotide repeats.

Although CRC patients are currently eligible for treatment only based on IHC results without molecular confirmation, a high failure rate of MMR-IHC in MSI diagnosis has been reported [16,21,22,23,24,25]. The disagreement between IHC and PCR could be attributable to various reasons related to both technical and biological issues which lead to misinterpretations. Our study showed a rate of discordant results between IHC and molecular approaches of 3.38% in CRC series, in line with previous studies [16,21,22,23,24,25]. The loss of MSH2 and MLH1 leads necessarily to MSI-H status since they are the primary partners of heterodimers, and unexpectedly our results showed that also the loss/patchy expression of the secondary partners, specificallyPMS2, could be associated with the MSI-H phenotype.

Among all MMR proteins analyzed, PMS2 was the most frequently lost or patchy, approximately in 80% of our CRC cases, in line with the the frequencies previously reported in the literature [23,31,32,33]. Furthermore, the isolated loss/patchy of PMS2 was associated with MSI-H phenotype in 8 (26.7%) out of 30 cases, regardless of the preserved expression of the primary partners.

Previous data showed that the isolated loss of PMS2 is found in about 4% of tumors with MSI phenotype [35]. Furthermore, the isolated loss of PMS2 has been described in MSI-H tumors carrying somatic gene variants [36]. In our CRC series, cases carrying lo-paMMR-IHC showed heterogeneous PCR results, and specifically 13 out of 46 cases were MSI-H, while all the others were MSS. Our findings suggest that lo-paMMR-IHC represents an equivocal subset that does not correspond to a definite molecular status ranging from MSS to MSI-H; therefore, they must necessarily be molecularly tested.

Beyond biological reasons, both technical limits and interpretative issues may affect the MMR-IHC results.

The technical limits resulting in false-negative staining of MMRs could be mainly linked to pre-analytical factors, especially tissue fixation [31,37]. Moreover, the misinterpretation could be due also to alternative staining patterns, such as cytoplasmic, dot-like or perinuclear staining.

Previous studies showed that a misdiagnosis of dMMR-IHC can be explained in about 20% of cases by a misinterpretation of staining or low technical performance leading to the absence of nuclear staining. [38,39].

MMR-IHC interpretation may be also subjective, mainly due to a lack of experience in the evaluation of such stainings.

The IHC evaluation is crucial in the clinical practice to define MSI status. Chen W and colleagues focused on the practical issues regarding MMR-IHC reporting and interpretation, paying particular care to the cutoff for normal staining, the staining variability/patchiness, the staining intensity of the tumor weaker than the control, the staining pattern post neoadjuvant therapy and the cytoplasmatic staining potentially associated with EPCAM-MSH2 fusion [39].

In this complex setting, our findings showed that the second opinion resolved the discrepancy between IHC and PCR in 3 out of 15 cases with discordant results, suggesting that a revision may improve the reclassification of cases with confounding stainings.

Thus, MMR-IHC could be used as a screening test; however, molecular testing is strongly recommended in cases with indeterminate IHC results, including for lo-paMMR and negative cases.

Noteworthy, our study showed that, unexpectedly, 2 out of 15 CRCs dMMR-IHC cases’ MSI status were not confirmed by PCR, suggesting that an additional molecular test could be necessary also in cases carrying a negative heterodimer to avoid false results. The therapeutic impact of the misdiagnosis of MMR-IHC could be dramatic. In our cohort of CRCs, the IHC results without molecular confirmation could have led to the wrong treatment with ICI in approximately 3.28% of patients carrying false dMMR IHC.

In these cases, both the biological reasons and the technical limitations associated with the microsatellite instability detection must necessarily be known to avoid the improper management of oncological patients. Previous studies reported the clinical impact of the misdiagnosis of MSI status defined through IHC led to ineffective treatment with ICI.

Despite high rates of response to ICI in MSI-H CRC patients, a high number of cases, ranging between 10% and 40%, exhibited primary resistance to the treatment. Beyond the biological mechanisms that could underlie the primary resistance to ICI of metastatic MSI-H CRC, the worrying point is that almost 10% of cases were resistant because they were false dMMR IHC [40]. This percentage is unacceptable since it is due to a misinterpretation of dMMR IHC, mainly because of evaluation by inexperienced pathologists. However, the use of both MMR-IHC and PCR is recommended to avoid the eligibility of the false dMMR for the treatment.

Previous studies have mainly focused on the determination of MSI in CRC, few data have currently been reported about MSI detection in GC. To the best of our knowledge, this is the first analysis based on the comparison between IHC and PCR and also in a series of GCs. Similar to the CRC series, in our cohort of GCs, only one case dMMR IHC was not MSI-H by PCR, whereas all cases with only PMS2 loss/patchy or with the heterodimer MLH1-PMS2 patchy were MSS by PCR. Although a disagreement between IHC and PCR was observed in our GC series, the discordance was low, approximately only 0.57%. The agreement between the MMR-IHC and MSI-PCR in our GC series results is strong (k = 0.957, 95% CI, *p* = 0.000). Our findings suggest that IHC screening is particularly reliable and conforms to the molecular test for MSI detection in GC, unlike the CRC. In clinical practice, MMR IHC still represents a valid screening method due to its low cost compared to molecular tests and its availability in most pathology units. Moreover, the molecular test is mandatory in doubtful cases.

The molecular assay currently used for MSI detection is mainly PCR, NGS represents another potential molecular approach, with the main advantage to determine simultaneously other targetable alterations and the tumor mutation burden, another molecular target for selection of the patients addressable to the immunotherapy. Moreover, NGS could solve the diagnosis in cases carrying MMR deficiency due to other mechanisms, including the somatic hypermethylation of the MLH1 gene promoter; an inherited germline mutation in one of the MMR genes (Lynch syndrome); and double somatic mutations in MMR genes [41,42]. However, the NGS approach could not always be feasible in small biopsies with a reduced amount of biomaterial available, and thus diagnosable through PCR.

Regarding the MSI detection on biopsy specimens, our findings showed total agreement with the corresponding surgical specimen, however, we analyzed a small series of biopsies. Similar to the assessment of other biomarkers for the pharmacognostic, the analysis of MSI status could be conditioned by the usual disadvantages linked to biopsy samples including the small amount of biomaterial available which could affect the subsequent molecular confirmation and the tumor heterogeneity, especially with MSH6 staining, as previously reported [16].

## 6. Conclusions

Finally, our findings suggest that in cases with loss/patchy MMR-IHC, the PCR could be mandatory to avoid misdiagnosis and consequently inadequate therapeutic choices. A feasible diagnostic flow chart to screen MSI status in daily clinical practice could include an additional molecular test exclusively for cases carrying loss or patchy MMR IHC.

## Figures and Tables

**Figure 1 cancers-14-02204-f001:**
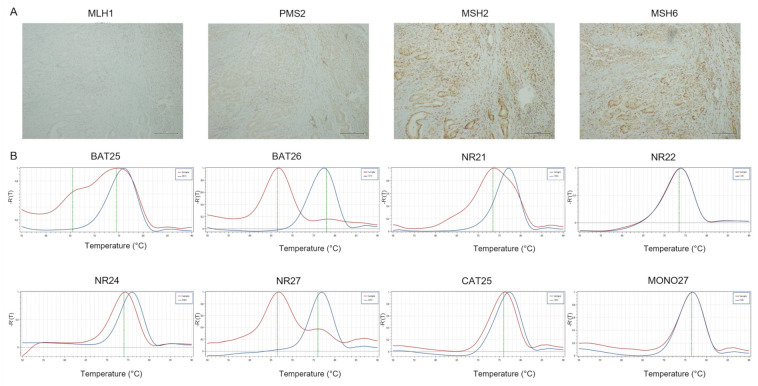
Representative results of CRC case showing dMMR-IHC and MSI-H by PCR. (**A**) MMR-IHC results: loss expression of MLH1 and PMS2 in tumor cells with positive internal control (original magnification 10×); intact expression of MSH2 and MSH6 in tumor cells (original magnification 10×). (**B**) MSI-PCR results: stability of NR22, NR24, CAT25 and MONO27; instability of BAT25, BAT26, NR21 and NR27 (red lines indicate samples while blue lines indicate MSS controls).

**Figure 2 cancers-14-02204-f002:**
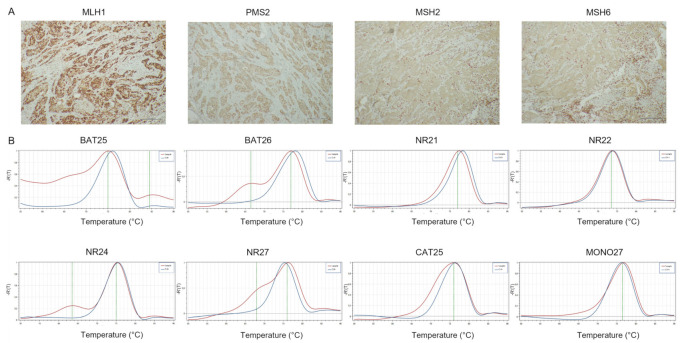
Representative results of CRC case showing dMMR-IHC and MSI-H by PCR. (**A**) MMR-IHC results: intact expression of MLH1 and PMS2 in tumor cells (original magnification 10×); loss expression of MSH2 and MSH6 in tumor cells with positive internal control (original magnification 10×). (**B**) MSI-PCR results: stability of NR21, NR22 and MONO27; instability of BAT25, BAT26, NR21, NR24, NR27 and CAT25 (red lines indicate samples while blue lines indicate MSS controls).

**Figure 3 cancers-14-02204-f003:**
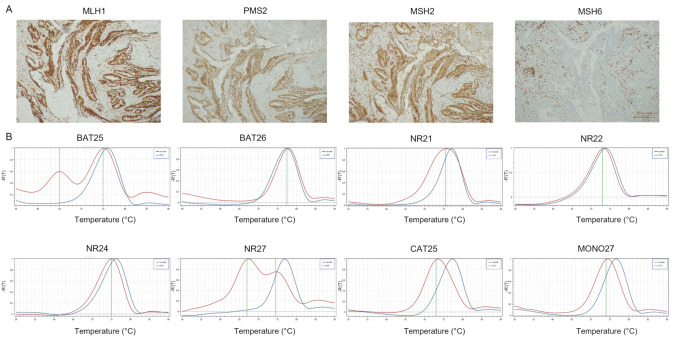
Representative results of CRC case showing loMMR-IHC and MSI-H by PCR. (**A**) MMR-IHC results: intact expression of MLH1, PMS2 and MSH2 in tumor cells (original magnification 10×); loss expression of MSH6 in tumor cells with positive internal control (original magnification 10×). (**B**) MSI-PCR results: stability of BAT26, NR22, NR24 and MONO27; instability of BAT25, NR21, NR27 and CAT25 (red lines indicate samples while blue lines indicate MSS controls).

**Figure 4 cancers-14-02204-f004:**
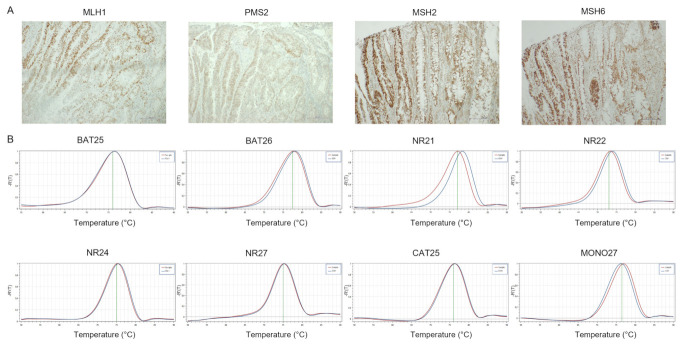
Representative results of CRC case showing paMMR-IHC and MSS by PCR. (**A**) MMR-IHC results: patchy expression of MLH1 and PMS2 in tumor cells with positive internal control (original magnification 10×); intact expression of MSH2 and MSH6 in tumor cells (original magnification 10×). (**B**) MSS-PCR results: stability of BAT25, BAT26, NR21, NR22, NR24, NR27, CAT25 and MONO27 (red lines indicate samples while blue lines indicate MSS controls).

**Figure 5 cancers-14-02204-f005:**
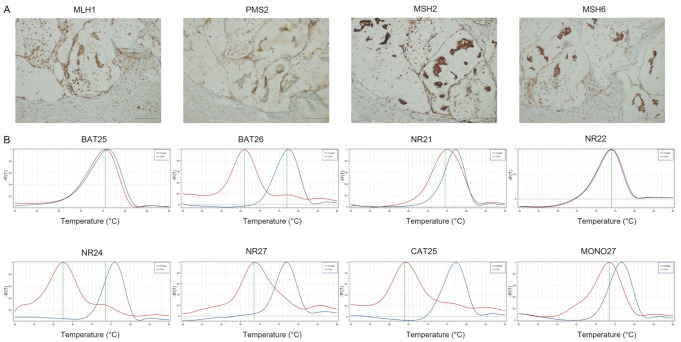
Representative results of CRC case showing paMMR-IHC and MSI-H by PCR. (**A**) MMR-IHC results: intact expression of MLH1, MSH2 and MSH6 in tumor cells (original magnification 10×); patchy expression of PMS2 in tumor cells with positive internal control (original magnification 10×). (**B**) MSI-PCR results: stability of BAT25 and NR22; instability of BAT26, NR21, NR24, NR27, CAT25 and MONO27 (red lines indicate samples while blue lines indicate MSS controls).

**Figure 6 cancers-14-02204-f006:**
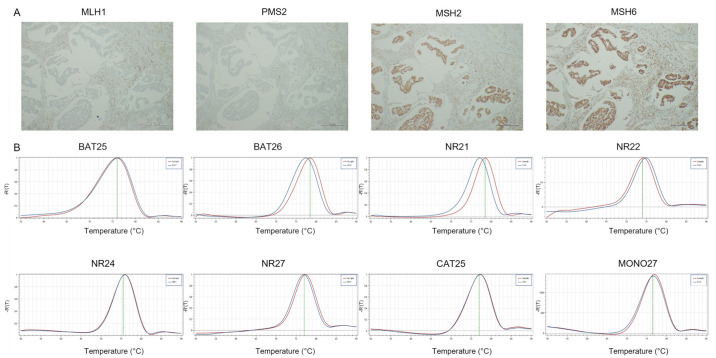
Representative results of GC case showing dMMR-IHC and MSS by PCR. (**A**) MMR-IHC results: loss expression of MLH1 and PMS2 in tumor cells with positive internal control (original magnification 10×); intact expression of MSH2 and MSH6 in tumor cells (original magnification 10×). (**B**) MSS-PCR results: stability of BAT25, BAT26, NR21, NR22, NR24, NR27, CAT25 and MONO27 (red lines indicate samples while blue lines indicate MSS controls).

**Figure 7 cancers-14-02204-f007:**
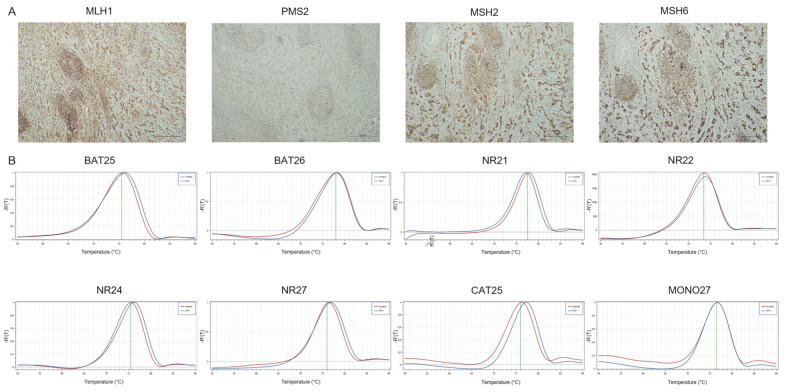
Representative results of GC case showing paMMR-IHC and MSS by PCR. (**A**) MMR-IHC results: intact expression of MLH1, MSH2 and MSH6 in tumor cells (original magnification 10×); patchy expression of PMS2 in tumor cells with positive internal control (original magnification 10×). (**B**) MSS-PCR results: stability of BAT25, BAT26, NR21, NR22, NR24, NR27, CAT25 and MONO27 (red lines indicate samples while blue lines indicate MSS controls).

**Table 1 cancers-14-02204-t001:** Clinical and pathological features of CRC patients.

Characteristics	N. Cases (%)	pMMR	dMMR	lo-paMMR
	444	383	15	46
**Age**				
<67	181 (40.76%)	156 (40.73%)	8 (53.33%)	17 (36.96%)
≥67	263 (59.24%)	227 (59.27%)	7 (46.67%)	29 (63.04%)
**Sex**				
male	265 (59.68%)	234 (61.1%)	9 (60%)	22 (47.83%)
female	179 (40.32%)	149 (38.9%)	6 (40%)	24 (52.17%)
**Tumor site**				
ascending colon	123 (27.71%)	91 (23.76%)	8 (53.33%)	24 (52.17%)
transverse colon	7 (1.57%)	4 (1.04%)	0	3 (6.52%)
descending colon	117 (26.36%)	114 (29.76%)	1 (6.67%)	2 (4.35%)
sigma-rectum	89 (20.04%)	79 (20.63%)	0	10 (21.74%)
metastatis	24 (5.40%)	20 (5.23%)	2 (13.33%)	2 (4.35%)
NA	84 (18.92%)	75 (19.58%)	4 (26.67%)	5 (10.87%)
**Histological type**				
adenocarcinoma	407 (91.67%)	366 (95.56%)	11 (73.33%)	30 (65.22%)
mucinous	33 (7.21%)	17 (4.44%)	3 (20%)	13 (28.26%)
signet ring cell	4 (1.12%)	0	1 (6.67%)	3 (6.52%)
**Grading**				
G1	24 (5.40%)	24 (5.40%)	0	0
G2	246 (55.41%)	240 (62.66%)	6 (40%)	31 (67.39%)
G3	76 (17.12%)	72 (18.80%)	4 (26.67%)	9 (19.57%)
NA	98 (22.07%)	93 (24.28%)	5 (33.33%)	6 (13.04%)
**B-RAF status**				
wt	326 (73.42%)	298 (77.80%)	4 (26.67%)	24 (52.17%)
mut	26 (5.86%)	12 (3.14%)	5 (33.33%)	9 (19.57%)
NA	92 (20.72%)	73 (19.06%)	6 (40%)	13 (28.26%)
**K-RAS status**				
wt	199 (44.82%)	170 (44.39%)	7 (46.67%)	22 (47.83%)
mut	153 (34.46%)	140 (36.55%)	2 (13.33%)	11 (23.91%)
NA	92 (20.72%)	73 (19.06%)	6 (40%)	13 (28.26%)
**N-RAS status**				
wt	338 (76.13%)	296 (77.29%)	9 (60%)	33 (71.74%)
mut	14 (3.15%)	14 (3.65%)	0	0
NA	92 (20.72%)	73 (19.06%)	6 (40%)	13 (28.26%)

NA: not available; wt: wild type; pMMR: proficient mismatch repair; dMMR: deficient mismatch repair; lo-paMMR: loss one and patchy mismatch repair; BRAF: v-raf murine sarcoma viral oncogene homolog B; KRAS: Kirsten rat sarcoma 2 viral oncogene homolog; NRAS: neuroblastoma rat sarcoma 2 viral oncogene homolog.

**Table 2 cancers-14-02204-t002:** Comparison between MMR-IHC and RT-PCR results in our CRCs series.

dMMR-IHC		MSI-PCR		
		MSI-H	MSI-L	MSS
**15 dMMR**		13		2
MLH1-PMS2	14	12	0	2
MSH2-MSH6	1	1	0	0
**46** **lo-paMMR**		13	3	30
**11 loss/1 MMR**				
MLH1	-	-	-	-
PMS2	9	6	-	3
MSH2	-	-	-	-
MSH6	2	2	-	-
**31 patchy**				
MLH1-PMS2	6	1	-	5
MSH2-MSH6	1	-	1	-
MLH1	-	-	-	-
PMS2	23	2	2	19
MSH2	1	-	-	1
MSH6	-	-	-	-
**4 loss/1 MMR patchy**				
MLH1-PMS2 P and MSH6 lo	1	-	-	1
MLH1 P and PMS2 lo	3	2	-	1

dMMR: deficient mismatch repair; IHC: immunohistochemistry; lo-paMMR: loss one and patchy mismatch repair; MLH1: MutL homolog human 1; MSH2: MutS homolog human 2; MSH6: MutS homolog human 6; PMS2: postmeiotic segregation increased 2; MSI-H: microsatellite instability-high; MSS: microsatellite stability; MSI-L: microsatellite instability-low.

**Table 3 cancers-14-02204-t003:** Second diagnostic opinion of MMR-IHC analysis and comparison with MSI-PCR results in CRC cases.

Cases	MMR-IHC I OPINION	MMR-IHC II OPINION	MSI-PCR	Concordance of MMR-IHC I/II Opinion with PCR
**1**	PMS2 neg	MLH1-PMS2 neg	MSI-H	II
**2**	MLH1 patchy-PMS2 neg	MLH1-PMS2 neg	MSI-H	II
**3**	MLH1-PMS2 patchy	MLH1-PMS2 neg	MSS	I
**4**	PMS2 patchy	MLH1-PMS2 patchy	MSI-H	no
**5**	PMS2 patchy	PMS2 neg	MSS	I/II
**6**	PMS2 neg	PMS2 pos	MSS	I/II

MMR: mismatch repair; IHC: immunohistochemistry; MLH1: MutL homolog human 1; PMS2: postmeiotic segregation increased 2; MSI-H: microsatellite instability-high; MSS: microsatellite stability.

**Table 4 cancers-14-02204-t004:** Clinical and pathological features of GC patients.

Characteristics	N Cases (%)	pMMR	dMMR	lo-paMMR
	176	157	13	6
**Age**				
<65	95 (53.98%)	92 (58.60%)	3 (23.08%)	0
≥65	81 (46.02%)	65 (41.40%)	10 (76.92%)	6 (100%)
**Sex**				
male	117 (66.48%)	107 (68.15%)	7 (53.85%)	3 (50%)
female	59 (33.52%)	50 (31.85%)	6 (46.15%)	3 (50%)
**Tumor site**				
gastro-esophageal	66 (37.49%)	62 (39.49%)	2 (15.38%)	2 (33.33%)
junction				
antrum-body	33 (18.19%)	31 (19.74%)	1 (7.69%)	0
body	61 (34.66%)	51 (32.49%)	9 (69.24%)	1 (16.67%)
metastasis	4 (2.27%)	4 (2.55%)	0	0
NA	13 (7.39%)	9 (5.73%)	1 (7.69%)	3 (50%)
**Lauren typing**				
intestinal type	124 (70.45%)	107 (68.15%)	12 (92.31%)	5 (83.33%)
diffuse type	45 (25.57%)	43 (27.39%)	1 (7.69%)	1 (16.67%)
mixed type	7 (3.98%)	7 (4.46%)	0	0
**Grading**				
G1	8 (4.54%)	7 (4.46%)	1 (7.69%)	0
G2	32 (18.18%)	29 (18.47%)	3 (23.07%)	0
G3	90 (51.14%)	77 (49.04%)	9 (69.24%)	4 (66.66%)
NA	46 (26.14%)	44 (28.03%)	0	2 (33.34%)
**HER2 status**				
negative	135 (76.70%)	120 (76.43%)	10 (76.92%)	5 (83.33%)
positive	23 (13.07%)	23 (14.65%)	0	0
NA	18 (10.23%)	14 (8.92%)	3 (23.08%)	1 (16.67%)
**EBV status**				
negative	135 (76.70%)	119 (75.80%)	13 (100%)	3 (50%)
positive	1 (0.57%)	0	0	1 (16.67%)
NA	40 (22.73%)	38 (24.20%)	0	2 (33.33%)
**PD-L1 status**				
negative	97 (55.11%)	90 (57.32%)	4 (30.77%)	3 (50%)
positive	48 (27.27%)	42 (26.75%)	4 (30.77%)	2 (33.33%)
NA	31 (17.62%)	25 (15.93%)	5 (38.46%)	1 (16.67%)

NA: not available; pMMR: proficient mismatch repair; dMMR: deficient mismatch repair; lo-paMMR: loss one and patchy mismatch repair; HER2: Erb-B2 receptor tyrosine kinase 2; EBV: Epstein–Barr virus; PD-L1: programmed death-ligand 1.

**Table 5 cancers-14-02204-t005:** Comparison between MMR-IHC and RT-PCR results in our GCs series.

dMMR-IHC		MSI-PCR		
		MSI-H	MSI-L	MSS
**13 dMMR**				
MLH1-PMS2	13	12	-	1
MSH2-MSH6	-	-	-	-
**6 lo-paMMR**				
**1 loss/1 MMR**				
MLH1	-	-	-	-
PMS2	1	-	-	1
MSH2	-	-	-	-
MSH6	-	-	-	-
**5 patchy**				
MLH1-PMS2	2	-	-	2
MSH2-MSH6	-	-	-	-
MLH1	-	-	-	-
PMS2	3	-	1	2
MSH2	-	-	-	-
MSH6	-	-	-	-

dMMR: deficient mismatch repair; pMMR: proficient mismatch repair; IHC: immunohistochemistry; lo-paMMR: loss one and patchy mismatch repair; MLH1: MutL homolog human 1; MSH2: MutS homolog human 2; MSH6: MutS homolog human 6; PMS2: postmeiotic segregation increased 2; MSI-H: microsatellite instability-high; MSS: microsatellite stability; MSI-L: microsatellite instability-low.

## Data Availability

The data presented in this study are available on request from the corresponding author.

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
