# Peer review of "Microsatellite Status Detection in Gastrointestinal Cancers: PCR/NGS Is Mandatory in Negative/Patchy MMR Immunohistochemistry"

_cancers, 2022, doi:10.3390/cancers14092204_

Round 1

Reviewer 1 Report

Dear author,

The work entitled "Microsatellite status detection in gastrointestinal cancers: molecular testing is mandatory in negative/patchy MMR immunohistochemistry" is great work but the results confirmed other data reported in literature: ESMO 2021 "Discordance between immunochemistry of mismatch repair proteins and molecular testing of microsatellite instability in colorectal cancer for example", or Malapelle et al 2020. For this reason there is no scientific novelty.

Moreover there is an important point to argue on:

1-Institutional ethical approval was not described for the performance of this study. The authors shoud insert Ethical Approval Number .

Best regards

Author Response

Reviewer 1

The work entitled "Microsatellite status detection in gastrointestinal cancers: molecular testing is mandatory in negative/patchy MMR immunohistochemistry" is great work but the results confirmed other data reported in literature: ESMO 2021 "Discordance between immunochemistry of mismatch repair proteins and molecular testing of microsatellite instability in colorectal cancer for example", or Malapelle et al 2020. For this reason there is no scientific novelty.

As correctly observed by the referee, previous studies have already reported the discordance between MMR IHC and MSI PCR in colorectal cancer, however, few data have been reported about MSI testing in gastric cancer. Until now, two main studies have analyzed the microsatellite status in gastric cancer (Ratti M, et al. Cell Mol Life Sci. 2018; Sugimoto R, et al Digestion. 2021), however no comparison between IHC and MSI molecular analysis was performed in GC series. Our study adressed the appropriate detection of MSI also in gastric cancer to avoid misinterpretations, to the best of our knolewdge, this the first study that compares different assays for MSI testing in GC.

Moreover there is an important point to argue on: 1-Institutional ethical approval was not described for the performance of this study. The authors shoud insert Ethical Approval Number.

Thank you the reviewer. We have inserted in the main text the Ethical Approval Number, as follow: “Institutional Review Board Statement: The study was approved by the Ethics Committee of University (iCURE protocol code: 790, 12 Dic 2018).”

Reviewer 2 Report

Main comments:
1) Please validate these findings using the TCGA colorectal adenocarcinoma and gastric adenocarcinoma datasets, respectively.
2) Was PD-L1 staining (+ve or -ve status) available for the CRC patients? It would be worthy to associate PD-L1 status with dMMR/MSI-H and compare it against pMMR/MSS tumor samples.

Minor comments:
1) How were 2 cases MSS (apart from 13 MSI-H) among 15 dMMR cases?
2) Please increase the font size in Figures 1-7. Alternatively, please opt to present some Figures as supplementary, as they seem to be providing repeating information. 
3) The correct terminology should be "instability", not "unstability".
4) A few comments on the MSI-PCR peaks:
 - Is MONO27 indeed stable according to the shown MSI-PCR peaks in Fig. 3B?
 - Is NR21 indeed stable according to the shown MSI-PCR peaks in Fig. 4B?
 - Are BAT26 and NR21 indeed stable according to the shown MSI-PCR peaks in Fig. 6B?
 - Is CAT25 indeed stable according to the shown MSI-PCR peaks in Fig. 7B?
5) line 311: please correct the sentence, as follows "however the final IHC diagnosis of lo-paMMR did not change"

Author Response

Reviewer 2

Main comments:
1) Please validate these findings using the TCGA colorectal adenocarcinoma and gastric adenocarcinoma datasets, respectively. 

Thank you the reviewer for this suggestion.

We compared our data of MMR-IHC alteration with the TCGA colorectal adenocarcinoma dataset: we observed an indicatively comparable percentage, particularly 13.7% of our series versus 15% of TCGA dataset. This small discrepancy can be justified since the TCGA includes in the not only alterations of MLH1, MSH2, MSH6 and PMS2, analyzed in our study, but also the inactivation of the MMR genes through aberrant promoter hypermethylation not analyzed in the present manuscript. Finally, we have implemented the text by adding in the CRC results: “In our serie MMR-IHC alteration was observed in about 13.7% of the cases analyzed, indicatively in accordance with the percentage reported in The Cancer Genome Atlas (TCGA) and consensus molecular subtype (CMS). [Inamura K. 2018; cancers]”

Similarly, we compared our data of MMR-IHC alteration with the TCGA gastric adenocarcinoma dataset: we observed 10.8% of our series versus 21% of TCGA dataset. This large discrepancy could be justified since the MSI GC subgroup is mostly due to promoter methylation of MLH1 rather than the mutations of MLH1, MSH2, MSH6 and PMS2, analyzed in our study. Thus, we have implemented the text by adding in the CRC results: “In our serie MMR-IHC alteration was observed in about 10.8% of the cases analyzed compared to 21% of MSI frequency reported in TCGA dataset. [Wang Q, Gastroenterology Res. 2019] This discrepancy could be justified since the MSI GC subgroup in TCGA classification is mostly due to promoter methylation of MLH1 rather than the mutations of MLH1, MSH2, MSH6 and PMS2, analyzed in our study.”

2) Was PD-L1 staining (+ve or -ve status) available for the CRC patients? It would be worthy to associate PD-L1 status with dMMR/MSI-H and compare it against pMMR/MSS tumor samples. 

Thank you the reviewer, the association between the PD-L1 status and dMMR/MSI-H is particularly interesting. We have performed PD-L1 IHC on all 61 cases carrying MSI dMMR-IHC and lo-paMMR-IHC, furthermore a control series of 60 cases pMMR also was analyzed for PD-L1 IHC expression. All data related to the PD-L1 status of CRC cases with dMMR-IHC and lo-paMMR are summarized in the Supplementary Table 1. Moreover, in the main text we added the follow sentence in the CRC results:

Moreover, all 61 cases harbouring dMMR or lo-paMMR were analyzed by PD-L1 IHC. Among 61 cases, 7  cases were PD-L1 positive, particularly 2 cases dMMR, 4 cases with PMS2 negative and 1 case MLH1-PMS2 patchy. All data are summarized in the Table S1. Furthermore, a series of 60 control cases determined as pMMR by IHC was analyzed by PD-L1 IHC, all cases were negative for PD-L1 expression.”

Minor comments:
1) How were 2 cases MSS (apart from 13 MSI-H) among 15 dMMR cases?

The 2 cases dMMRIHC but MSS-PCR represent an unexpected result of our study. These two cases showed either the same heterodimer altered MLH1-PMS2, this is a critical point that could suggest another biological reason associated with the loss of these MMR. Probably, the loss of  MLH1-PMS2 could be explained through the aberrations not detected through PCR. However, NGS results confirmed for either case the MSS status. We have improved the main text adding in the results the following sentence: “The 2 cases dMMR-IHC and MSS-PCR showed the loss of the same heterodimer MLH1-PMS2.”

2) Please increase the font size in Figures 1-7. Alternatively, please opt to present some Figures as supplementary, as they seem to be providing repeating information. 

As correctly observed by the reviewer, we changed the Figure1-7 improving the font size and also we improved the font sizes of the axes on the graphs. We have inserted the new figures in the main text.  

3) The correct terminology should be "instability", not "unstability".

Thank you the reviewer for this suggestion. We changed the wrong term "unstability" in "instability" and the changes are marked red in the main text. 

4) A few comments on the MSI-PCR peaks:
 - Is MONO27 indeed stable according to the shown MSI-PCR peaks in Fig. 3B?
 - Is NR21 indeed stable according to the shown MSI-PCR peaks in Fig. 4B?
 - Are BAT26 and NR21 indeed stable according to the shown MSI-PCR peaks in Fig. 6B?
 - Is CAT25 indeed stable according to the shown MSI-PCR peaks in Fig. 7B?

We have reviewed all MSI-PCR peaks and we confirm that they are correct. In particular, the curves of the samples do not necessarily have to coincide perfectly with the curve of the stable control, as shown in the figures reported in the datasheet of the kit EasyPGX ready MSI. (https://www.diatechpharmacogenetics.com/).

The sample is defined as stable on the basis of the values of ΔTm, regardless of the perfect superposition of the two curves, according to the manufacturer's instructions.

5) line 311: please correct the sentence, as follows "however the final IHC diagnosis of lo-paMMR did not change"

Thank you the reviewer for this suggestion. As correctly suggested by the reviewer, we changed the sentence as follows:

“In the remaining 3 cases, although the interpretation of the second pathologist was discordant with the first regarding the definition of one MMR immunohistochemical staining, however the second opinion did not overall change the diagnosis of the cases which were confirmed as lo-paMMR (Table 3).”

Reviewer 3 Report

The authors report two series of GI cancers: colorectal and gastric cancers in which MMR system was assessed using IHC of the four proteins. The results were compared with PCR testing of  microsatellite instability.

Introduction is lacking some references. Especially the use of ICI in MSI patients is supported by a review whereas several international Phase III trials could be cited.

Presentation of the results includes examples of stainings and the corresponding PCR results which should be appealing for non specialists in the field. Individual results are available in supplementary materials.

Patients have been included from 2019 to 2021. This short period of inclusion guarantees homogeneity in the techniques applied to the collected samples. Still, 20% of recommended molecular biology analyses are missing. PD-L1 and , above all, HER2 status, are missing in 30% of cases in the series of gastric cancers which is high.

The authors underline  the high rate of non contributive NGS. Very few cases have been analyzed by NGS. It is thus hard to draw any conclusion from these analyses. It would be of great interest to get more samples analysed by NGS with TMB data.

Some passages look fast written. For instance, line 401, while the authors are reporting the GC series, they refer to "CRC biopsy specimens"

In both gastric and colorectal cancer series, the authors refer to a control series of 100 and 150 control cases respectively. Is it from a different series or from the original larger ones presented in the study? What could justify to analyse only part of the series without introducing a bias? Although it could make the manuscript heavier, the characteristics of these selected pMMR cases should be presented at least in supplementary material.

The Discussion section requires some more writing and references.

The authors detail the impact of MSI status misdiagnosis which is rather related to the introduction section as it justifies the study.

Then, the authors state false dMMR IHC are due to un-experienced pathologists without any citation supporting this conclusion. The related paragraphs are more judgemental than scientific. Instead, the authors could elaborate further about the relevance of PMS2 IHC.

Author Response

Reviewer 3

The authors report two series of GI cancers: colorectal and gastric cancers in which MMR system was assessed using IHC of the four proteins. The results were compared with PCR testing of microsatellite instability.

  • Introduction is lacking some references. Especially the use of ICI in MSI patients is supported by a review whereas several international Phase III trials could be cited.

Thank you the reviewer for his suggestion that improved our manuscript. As correctly observed by the reviewer, in the introduction some references were missing and the ICI use in patients harboring MSI-H/dMMR was not adequately emphasized. In this view, we improved the Introduction by adding the following sentences and references:

“Recently, the KEYNOTE-177 phase III trial have demonstrated the superiority of pembrolizumab in treatment-naive mCRC over first-line chemotherapy with or without biological agents in terms of progression-free survival (PFS), overall response rate (ORR) and safety. [André T, N Engl J Med. 2020]”

“Furthermore, post hoc analyses of randomized controlled trials (RCTs) suggested the superior efficacy of anti-PD-1-based therapy compared with standard chemotherapy in MSI-high GC patients. [Pietrantonio F, ESMO Open. 2021] [Shitara K, Lancet. 2018] [Shitara K, Ann Oncol. 2019] [Shitara K, JAMA Oncol. 2020] [Chao J, J Clin Oncol. 2020;] [Moehler M, Ann Oncol. 2020].  Based on these data, the immunotherapy could be routinely used as the treatment of choice for MSI-high CRCs and GCs patients.”

Presentation of the results includes examples of stainings and the corresponding PCR results which should be appealing for non specialists in the field. Individual results are available in supplementary materials.

  • Patients have been included from 2019 to 2021. This short period of inclusion guarantees homogeneity in the techniques applied to the collected samples. Still, 20% of recommended molecular biology analyses are missing. PD-L1 and , above all, HER2 status, are missing in 30% of cases in the series of gastric cancers which is high.

Unfortunately, the amount of biomaterial was not always available to perform both PDL1 and HER analysis. We re-evaluated all GC cases of our series and we were able to integrate the following data: 35 new cases were analyzed for HER2 and 20 new cases for PD-L1. We have modified the data both in the paragraph of results and table 4 “clinical and pathological features of GC patients”. 

The authors underline  the high rate of non contributive NGS. Very few cases have been analyzed by NGS. It is thus hard to draw any conclusion from these analyses. It would be of great interest to get more samples analysed by NGS with TMB data. 

As correctly observed by the referee, we had analyzed too few cases by NGS. We have increased the number of cases analyzed by NGS. In particular, we selected cases that showed lo-paMMR staining and cases with discordant results between IHC and PCR in order to clarify the MSI status using another molecular assay.

We have increased the NGS test in other 16 CRC cases, we changed the Table S3 and we improved the main text as follow:“Moreover, 25 out of 444 cases were further analyzed by NGS, all cases had contributory results, particularly: 7 cases pMMR-IHC and MSS-PCR confirmed MSS status by NGS; 2 cases dMMR-IHC and MSS-PCR showed MSS status by NGS; 8 cases loMMR-IHC and MSI-H-PCR showed MSI-H status by NGS; 3 cases paMMR-IHC and MSI-H-PCR showed MSI-H status by NGS;  3 cases paMMR-IHC and MSI-L-PCR showed MSS status by NGS;  2 cases lo-paMMR-IHC and MSI-H-PCR showed MSI-H status by NGS. All data are summarized in Table S3.”

We have increased the NGS test in other 2 GC cases, we changed the Table S5 and we improved the main text as follow: “Moreover, 9 out of 176 cases were further analyzed by NGS, however 4 out of 9 cases not had contributory results due to an adequate biomaterial to NGS analysis. All 3 cases dMMR-IHC and MSI-H by PCR confirmed the instability by NGS, the case MLH1-PMS2 negative and MSS-PCR confirmed MSS status by NGS  and the case PMS2 patchy and MSI-L-PCR showed MSS status by NGS. All data are summarized in Table S5.”

Some passages look fast written. For instance, line 401, while the authors are reporting the GC series, they refer to "CRC biopsy specimens"

Thank you the reviewer for this suggestion. We changed the mistake in the main text.

In both gastric and colorectal cancer series, the authors refer to a control series of 100 and 150 control cases respectively. Is it from a different series or from the original larger ones presented in the study? What could justify to analyse only part of the series without introducing a bias? Although it could make the manuscript heavier, the characteristics of these selected pMMR cases should be presented at least in supplementary material.

As correctly observed by the referee, we have described the series of cases pMMR analyzed also by PCR in an unclear and confusing way. We did not use different control series, but rather we have selected some case pMMR IHC and we have analyzed these cases also through MSI molecular testing. We are sorry for the very confusing exposure, thus we have modified the main text as follow:

“We selected 150 out of 383 CRC pMMR IHC and we analyzed these cases by PCR, all cases confirmed MSS status.” ; “We selected 100 out of 157 GC pMMR IHC and we analyzed these cases by PCR, all cases confirmed MSS status.”

The Discussion section requires some more writing and references. The authors detail the impact of MSI status misdiagnosis which is rather related to the introduction section as it justifies the study.Then, the authors state false dMMR IHC are due to un-experienced pathologists without any citation supporting this conclusion. The related paragraphs are more judgemental than scientific.

As correctly observed by the referee, the discussion about MMR-IHC interpretation lacked adequate scientific support and related references. We have implemented this aspect and supported our data with other references, as follows:

“Previous studies showed that a misdiagnosis of dMMR-IHC can be explained in 20% of cases by an incorrect interpretation of staining or low technical performancelaeding to the absence of nuclear staining. [Chen W, Diagn Pathol. 2017] [Chen W, Mod Pathol. 2019;]”

“The IHC evaluation and reporting play a pivotal role in the clinical practice to define MSI status. Chen W and colleagues focused on the practical issues regarding MMR-IHC reporting and interpretation, paying particular care to the cutoff for normal staining, the staining variability/patchiness, the staining intensity of the tumor weaker than the control, the staining pattern post neoadjuvant therapy, the cytoplasmatic staining potentially associated to EPCAM-MSH2 fusion. [Chen W, Mod Pathol. 2019;]”

Instead, the authors could elaborate further about the relevance of PMS2 IHC. 

As correctly observed by the referee, the discussion about PMS2 IHC could be implemented,thus we implemented the main text as follows:

“Previous data showed that the isolated loss of PMS2 immunostaining is found in 4% of tumours with MSI phenotype. [Alpert L, Arch Pathol Lab Med 2018] Furthermore, the isolated loss of PMS2 has been described in MSI-H tumors carrying somatic gene variants. [Stelloo E, Ann Oncol 2017] “

Reviewer 4 Report

The comparison between different methods of MSI detection is valuable for colorectal cancer diagnosis. It would be helpful if the authors could address the following concerns.

  1. Could the authors include information about MSH3 mutations in MSI? Since MSH3 is important for MMR and there is some evidence for its involvement in MSI, it would be helpful for the readers if the authors provide their rationale for not including it in their analysis. It would make the introduction more robust.
  2. It would be beneficial for the readers if the authors could improve the font sizes of the axes and legends on the graphs? This would help the readers with the interpretations of the graphs.

Author Response

Reviewer 4

The comparison between different methods of MSI detection is valuable for colorectal cancer diagnosis. It would be helpful if the authors could address the following concerns.

  1. Could the authors include information about MSH3 mutations in MSI? Since MSH3 is important for MMR and there is some evidence for its involvement in MSI, it would be helpful for the readers if the authors provide their rationale for not including it in their analysis. It would make the introduction more robust.

The MSH3 mutations in MSI field represent a very interesting aspect. Although the clinical value of the MSH3 mutations will be a next potential therapeutic target, however, it is not the issue of the present study since the main aim is the comparison between different assays to MSI detection. Particularly, neither IHC nor PCR analyze the MSH3 mutations, thus this aberration is not contributive to our analysis.

  1. It would be beneficial for the readers if the authors could improve the font sizes of the axes and legends on the graphs? This would help the readers with the interpretations of the graphs.

As correctly observed by the reviewer, we changed the Figure1-7 improving the font size and also we improved the font sizes of the axes on the graphs. We have inserted the new figures in the main text.  

Reviewer 5 Report

Concordance between MSI status determine by IHC or PCR has already widely been described. The conclusion of the article for MSI testing are ESMO recommendations since 2019.

Of note, to validate a screening test, it is mandatory to know the rate of false negative. In this study, the authors did not compare IHC and PCR for pMMR patients, thus this rate is undetermined and then can not support the conclusion.

Author Response

Reviewer  5

Concordance between MSI status determine by IHC or PCR has already widely been described. The conclusion of the article for MSI testing are ESMO recommendations since 2019.

As rightly observed by the reviewer, the conclusions of our manuscript overlap with guidelines proposed by ESMO in 2019. Furthermore, previous studies have already reported the discordance between MMR IHC and MSI PCR in colorectal cancer, however, few data have been reported about MSI testing in gastric cancer. Until now, two main studies have analyzed the microsatellite status in gastric cancer (Ratti M, et al. Cell Mol Life Sci. 2018; Sugimoto R, et al Digestion. 2021), however, no comparison between IHC and MSI molecular analysis was performed in GC series. Our study addressed some novelty issues, including i) the appropriate detection of MSI also in gastric cancer to avoid misinterpretations, to the best of our knowledge, this is the first study that compares different assays for MSI testing in GC ii) the use of NGS assay to resolve cases with discordant results between IHC and PCR in both CRC and GC. We hope that the reviewer will appreciate these aspects of this manuscript and the revisions made to implement it.

Of note, to validate a screening test, it is mandatory to know the rate of false negative. In this study, the authors did not compare IHC and PCR for pMMR patients, thus this rate is undetermined and then can not support the conclusion.  

As correctly suggested by the reviewer, in our study the definition of the rate of false-negative is mandatory. Unfortunately, PCR was not performed in all cases since the series analyzed were really very numerous. However, several pMMR cases were analyzed by PCR, particularly 150 out of 383 CRC pMMR and 100 out of 157 GC pMMR. In this view, we have assumed as the gold standard assay the PCR test and we calculated the specificity and sensibility of IHC to detect MSI status based on the CRC and GC cases analyzed both by IHC and PCR. In CRC series, IHC showed a sensibility of 50% and a specificity of 99%. IN GC series, IHC showed a sensibility of 100% and a specificity of 99%.  We added the IHC sensibility and specificity in the text in the Results of CRC and GC respectively, as follow: “Considering all cases of our series analyzed both by IHC and PCR, the IHC showed a  sensibility of 50% and a specificity of 99%” ; “Considering all cases of our series analyzed both by IHC and PCR, the IHC showed a  sensibility of 100% and a specificity of 99%”. We hope that the reviewer may agree with our proposal to resolve this lack and the revisions made to implement it. 

Round 2

Reviewer 1 Report

Dear author, 

you have completed the work, even if there is not an important novelty. Anyway there are an important numbers of cases. Please could you exactly insert the number and the name of your obsarvational protocol using the samples from patients?

Author Response

We thank the reviewer for the previous suggestions provided to us. We have checked the number and the name of our observational protocol using the samples from patients.

Reviewer 2 Report

1) Please edit the Introduction so as not to have any blank lines or long spaces between sentences (e.g., lines 60, 62, 66, 70, etc.)
2) In addition, moderate English changes are required. Please make an effort to apply the necessary edits needed language-wise throughout the text (e.g., replace "serie" with "series", "Immunohistochemistry analysis" with "Immunohistochemical analysis", etc).
3) Page 6, line 181: the "and consensus molecular subtype (CMS)" does not seem to fit in this sentence. Please erase and include the name of the TCGA dataset with which results were compared (TCGA-COADREAD). Similarly, on page 12, please mention the TCGA dataset name for GC.

Author Response

We thank the reviewer for the suggestions provided to us.

1) Please edit the Introduction so as not to have any blank lines or long spaces between sentences (e.g., lines 60, 62, 66, 70, etc.)

Done

2) In addition, moderate English changes are required. Please make an effort to apply the necessary edits needed language-wise throughout the text (e.g., replace "serie" with "series", "Immunohistochemistry analysis" with "Immunohistochemical analysis", etc).

Done

3) Page 6, line 181: the "and consensus molecular subtype (CMS)" does not seem to fit in this sentence. Please erase and include the name of the TCGA dataset with which results were compared (TCGA-COADREAD). Similarly, on page 12, please mention the TCGA dataset name for GC.

As correctly observed, we have entered the precise reference of the TCGA datasets. We thank the reviewer for his observation.

Reviewer 3 Report

All the points raised in the previous version were adressed.

Especially, the authors put a great effort on reducing the number of missing data and improving the number of NGS analyses.

Author Response

We thank the reviewer for the valuable suggestions provided to us.